# Coherent multidimensional spectroscopy of dilute gas-phase nanosystems

Lukas Bruder [1], Ulrich Bangert[1], Marcel Binz[1], Daniel Uhl[1], Romain Vexiau[2], Nadia Bouloufa-Maafa[2], Olivier Dulieu[2] & Frank Stienkemeier [1,3]

Two-dimensional electronic spectroscopy (2DES) is one of the most powerful spectroscopic techniques with unique sensitivity to couplings, coherence properties and real-time dynamics of a quantum system. While successfully applied to a variety of condensed phase samples, high precision experiments on isolated systems in the gas phase have been so far precluded by insufficient sensitivity. However, such experiments are essential for a precise understanding of fundamental mechanisms and to avoid misinterpretations. Here, we solve this issue by extending 2DES to isolated nanosystems in the gas phase prepared by helium nanodroplet isolation in a molecular beam-type experiment. This approach uniquely provides high flexibility in synthesizing tailored, quantum state-selected model systems of single and many-body character. In a model study of weakly-bound $Rb_2$ and $Rb_3$ molecules we demonstrate the method's unique capacity to elucidate interactions and dynamics in tailored quantum systems, thereby also bridging the gap to experiments in ultracold quantum science.

[1] Institute of Physics, University of Freiburg, 79104 Freiburg, Germany. [2] Laboratoire Aimé Cotton, CNRS, Université Paris-Sud, ENS Cachan, Université Paris-Saclay, 91405 Orsay Cedex, France. [3] Freiburg Institute of Advanced Studies (FRIAS), University of Freiburg, Albertstr. 19 79194 Freiburg, Germany. Correspondence and requests for materials should be addressed to L.B. (email: lukas.bruder@physik.uni-freiburg.de)

A key objective in science is to understand the elementary microscopic processes that drive nature. Addressing this question is—up to date—a major experimental and theoretical challenge, as molecular dynamics occur on ultrashort time scales (picoseconds to sub-femtoseconds) and involve the complex interplay of many degrees of freedom (for example electronic, rovibrational, structural, and environmental). The development of 2DES has considerably advanced this field as it has improved the time and frequency resolution of molecular dynamics to an unprecedented level[1]. This technique maps the system's third order nonlinear response onto 2D frequency-correlation maps which provide invaluable information over one-dimensional methods. Key advantages are the high spectro-temporal resolution, the direct disclosure of couplings and the differentiation of homogeneous and inhomogeneous broadening mechanisms[1–3]. As such, 2DES has provided insights in topics as broad as energy relaxation pathways and quantum coherence in photosynthetic systems[4–7], many-body correlations and exciton dissociation in semiconductor materials[8,9] and reaction pathways in photophysical/-chemical reactions[10].

Despite the success of 2DES, the vast complexity of investigated condensed phase systems makes precise analysis and modeling extremely difficult. This has led to some ambiguities in interpretations, most prominently the observation of long-lived quantum coherences in biological systems[11,12]. 2DES studies of single, isolated systems in the gas phase would strongly reduce the complexity and are therefore of immense interest. In addition, gas-phase studies provide access to highly selective observables that can provide crucial information not available in the condensed phase. For instance, information about dark states, ion-mass spectra or ion/electron angular distributions can be deduced[13,14].

A first demonstration of 2DES combined with mass-selected photoion detection has been recently reported[15], revealing details about ionization pathways in $NO_2$ prepared in a thermal molecular gas. This study unveiled the challenge of the extremely high sensitivity required for 2DES gas-phase experiments. As such, acquisition times and data quality were clearly below the level of condensed-phase experiments making precise analysis of the gas-phase data still difficult. Another drawback of gas-phase experiments is the constrained flexibility in the synthesis of samples. For instance, the preparation of molecular complexes/aggregates in the gas phase is technically very restricted. Consequently, the aspects of inter-particle couplings/dynamics and environmental effects, being the essence of most microscopic processes, cannot be modeled in such studies.

The latter issue can be solved with helium nanodroplet isolation (HENDI)[16]. In this approach, a supersonic beam of superfluid helium droplets is doped with single or multiple spectroscopic probes, forming an isolated well-controlled nanosystem in the gas phase. In recent years, HENDI has been established as a unique technique for spectroscopic studies of atoms, molecules and their complexes, that are isolated in the superfluid helium matrix[16]. Spectroscopy of pigment molecules[17], up to larger biomolecules[18], and exotic species[19,20] has been demonstrated with a resolution often clearly exceeding other methods[16]. Translational and internal degrees of freedom of embedded species are efficiently cooled to sub Kelvin (370 mK) temperatures[16]. Heterogeneous complexes and molecular aggregates are readily synthesized directly in the droplets[20–24]. Thereby, the rare-gas environment provides a prototypical perturbation which is much simpler to model than the influence of molecular solvent networks, and by co-doping with individual atoms/molecules (microsolvation)[23], environmental parameters can be tuned and controlled to a degree much higher than in most other experiments.

One distinct class of molecules which have been extensively studied in recent years with HENDI, are alkali-metal molecules in their weakly-bound high-spin states[21,25,26]. Here, the helium droplet surface serves as a cryogenic substrate to form the weakly-bound van der-Waals alkali-metal complexes, which are otherwise hard to access with spectroscopic methods. Particular interest has been devoted to these molecular species, as they play an important role in ultracold physics and chemistry[27] and serve as an ideal test bench for ab initio quantum chemistry methods[28–30]. They also provide intriguing model systems for moderate to strong couplings to the superfluid helium environment[25,26,31] and are thus ideal to demonstrate distinct aspects of system-bath interactions of single molecules coupled to an environment.

In the current work, we introduce the combination of 2DES with HENDI and investigate the coherent photoinduced dynamics and system-bath interactions of $Rb_2$ and $Rb_3$ molecules attached to the surface of helium nanodroplets. Our model study represents a significant progress in gas-phase 2DES, as it solves the previous problems of insufficient experimental sensitivity and lacking flexibility in the synthesis of gas-phase samples. As such, the presented approach uniquely enables 2DES studies of isolated, tailored model systems ideally suited to study intramolecular as well as intermolecular properties/dynamics and the influence of system-bath interactions while retaining the system's overall complexity at a level much smaller than in condensed phase systems.

## Results

**2DES of isolated nanosystems in the gas phase.** To demonstrate the various aspects of our method, $Rb_2$ and $Rb_3$ molecules are prepared in their high-spin electronic ground states directly on the surface of the helium droplets (Fig. 1a, details in Materials section). To this end, helium nanodroplets are formed in a supersonic jet expansion and doped with individual Rb atoms in a pick-up process. Thereby, the droplets serve as a cold, inert substrate assisting the molecule formation, the natural selection of their van der Waals-bound high-spin configurations $Rb_2$ $a^3\Sigma_u^+$ and $Rb_3$ $1^4A_2'$ and cooling to their lowest vibrational level. These states are otherwise difficult to access due to their low binding energy $E_B$ ($Rb_2$ $E_B = 235$ cm$^{-1}$,[29] $Rb_3$ $E_B = 939$ cm$^{-1}$[30]), which thus exemplifies the ability of HENDI for tailored molecular synthesis. While demonstrated here for the synthesis of alkali-metal molecules, in the same fashion aggregates of larger organic molecules have been readily formed and isolated[24]. Regarding the experimental technique and the required sensitivity, conditions are similar for the larger molecules/clusters, but for demonstrating high resolution at dilute conditions, simple molecular structures such as the studied $Rb_2$ and $Rb_3$ molecules are preferable. Note, that in contrast to all previous 2DES studies, here, the molecular samples are cooled to the sub-Kelvin regime, leading to a narrow initial quantum state distribution, which greatly simplifies the analysis of the 2DES data and allows us to deduce fine details of the molecular structure and dynamics.

A major challenge in gas-phase experiments are the low target densities, making the application of advanced nonlinear spectroscopy methods extremely demanding. Especially, in HENDI experiments, densities are only $\leq 10^7$ cm$^{-3}$ (corresponds to optical density (OD) of $\sim 10^{-11}$, see Supplementary Note 2). The routinely employed experimental implementation of 2DES based on non-collinear four-wave-mixing is not suitable for such low molecular densities. To achieve the required high sensitivity, we instead use a collinear geometry, rapid phase modulation combined with efficient lock-in detection[32] and photoionization for detection (Fig. 1b, details in Material section).

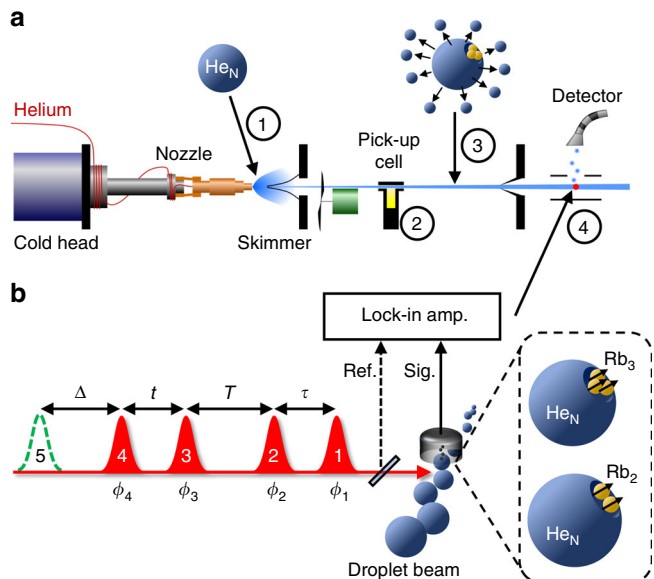

**Fig. 1** Experimental scheme. **a** Molecular beam-type vacuum apparatus for helium nanodroplet beam generation upon adiabatic expansion of helium (1), followed by doping with Rb atoms in a pick-up cell (2) and evaporative cooling of the formed Rb molecules (3). The isolated, cold $Rb_2$ and $Rb_3$ molecules intersect with the laser beam and photoelectrons/-ions are detected (4). **b** Four phase-modulated laser pulses with delays $\tau$, $T$ and $t$, excite and ionize the prepared molecules in the droplet beam. The phase $\phi_i$ of each pulse is individually modulated at kHz-frequencies, leading to a modulation-beat of the photoelectron/-ion yield. A lock-in amplifier is used for demodulation and isolation of the nonlinear 2D signal components. For a more selective ionization, a fifth pulse delayed by $\Delta$ is optionally applied

In this scheme, four phase-modulated laser pulses induce a fourth-order nonlinear population in the sample which is mapped onto the photoionization yield (Supplementary Fig. 3). The weak nonlinear signal contributions are extracted from detected photoelectron or mass-resolved photoions based on their individual phase modulation signatures. This procedure is similar to phase-cycling, however, is performed here at an update rate of 200 kHz, and is therefore more sensitive than most pulse shaper-based setups. Likewise, photoionization is of advantage as it ensures higher collection efficiencies than photon detection and allows for selective probing through different ionization channels.

**Photoionization-2DES of $Rb_2$ and $Rb_3$ attached to helium nanodroplets.** Photoelectron-2D spectra of the isolated molecular species are shown in Fig. 2a, b and an ion-detected 2D spectrum is shown in Fig. 2c. These 2D maps directly correlate the pump excitation ($\omega_\tau$-axis, comparable to absorption spectrum) with the system response ($\omega_t$-axis, comparable to emission spectrum), probed as a function of the evolution time $T$. Thereby, the encoded phase information enables clear discrimination of signal contributions: ground state bleach (GSB), stimulated emission (SE) both positive and excited state absorption (ESA) negative amplitude (details in Supplementary Note 1 and Supplementary Fig. 3). Furthermore, the peak magnitude strongly depends on the ionization scheme. This enables selective enhancement/discrimination of individual features (demonstrated in Fig. 2c), which, in contrast to previous 2DES experiments, provides us an additional means to disentangle the system response.

Considering the extremely low molecular densities in the experiment, the acquired 2D spectra reveal very high quality. The

sharp, well-separated spectral features allow us unambiguous identification of spectral components and correlations among those. Absorption and emission profiles of the data (pump/probe projections, Supplementary Fig. 4) are in excellent agreement with ab initio calculations[28,29] and high-resolution steady-state laser spectroscopy[19,21], which confirms the high fidelity of our method and facilitates clear assignment of all spectral features. At $\omega_\tau = 14125\,\text{cm}^{-1}$ we observe the $Rb_3$ $1^4A_2' \rightarrow 1^4A_{1,2}''$ quartet transition and correlated negative ESA peaks, as well as a transient positive cross peak ($T = 0$ fs). Around $\omega_\tau = 13500\,\text{cm}^{-1}$, we observe the $Rb_2$ $a^3\Sigma_u^+ \rightarrow (1)^3\Pi_g$ triplet resonance with clearly resolved spin-orbit (SO) components $0_g^\pm, 2_g$ of the excited state along the $\omega_\tau$-axis. Note, that the $0_g^\pm$ components are almost energetically degenerate[28] and in accordance with previous experiments[19], the $1_g$ component is not observed in HENDI experiments. Correlated to the $a^3\Sigma_u^+ \rightarrow (1)^3\Pi_g$ resonance, we identify two ESA and one cross peak at off-diagonal positions (labeled $ESA_1$, $ESA_2$, and CP). The high resolution allows us furthermore to identify distinct Stokes shifts (red shift of diagonal peaks along $\omega_\tau$-axis, see also Fig. 3a). This asymmetry in absorption and emission is characteristic for molecules initially prepared in a single vibrational ground state[33], confirming the preparation of cold molecules in our experiment.

As an intriguing aspect, we observe a persistent coherent oscillation of the $Rb_2$ ESA peaks as a function of the evolution time $T$ (Fig. 2d). As the beat appears in spectrally well-isolated ESA pathways, it unambiguously reflects a vibrational wave packet prepared in the $(1)^3\Pi_g, \Omega = 0_g^\pm, 2_g$ excited states (Fig. 2e). Franck-Condon (FC) calculations reveal a wave packet excitation around $v = 7$ with an average level spacing of $\approx 20\,\text{cm}^{-1}$ (Supplementary Fig. 5) in very good agreement with the observed oscillation period of $\approx 1550$ fs. The phase information contained in the 2D spectra in addition allows the identification of two separate FC windows. The initial phase of the $ESA_1$ and $ESA_2$ oscillations imply a FC window at the inner/outer turning points of the $(1)^3\Pi_g$ potential energy curves (PECs, Fig. 2e), respectively, which completes the picture of the $Rb_2$ wave packet dynamics and the molecule's ionization pathways.

Previously, the red-shifted emission observed at $\omega_\tau = 12950\,\text{cm}^{-1}$ (CP feature) has been assigned to a resonance in desorbed, free gas-phase $Rb_2$ molecules[19]. Interestingly, the extended information disclosed in our 2DES experiment indicates that this feature rather corresponds to a distinct interaction of the $Rb_2$ molecule with the helium environment. Our data reveals a beat of the CP feature in phase with the $ESA_2$ peak (not shown) and shows the same emission frequency for all SO components. This points to an ultrafast relaxation at the outer turning point of the $0_g^\pm, 2_g$ states to a common lower-lying state. Vibrational relaxation within these states would not reproduce the observed red shift. Since no other electronic states are in spectral vicinity, we can conclude, that the CP feature originates from a relaxation/tunneling into the outer potential well of the $0_g^+$ state[28], catalyzed by the perturbation of the helium environment. The electron distribution of this state differs from the other bound $(1)^3\Pi_g$ states which might lead to a slightly smaller $Rb_2$-$He_N$ interaction (Fig. 2e) and thus would explain the red-shifted emission. This behavior is qualitatively confirmed by a calculation of the $Rb_2$-He interaction potential for various points along the interaction coordinate, indicating a strong effect on the $Rb_2$ PECs as a function of the $Rb_2$-$He_N$ distance. Note, that in Fig. 2e only a schematic of this effect is shown. The CP feature thus reflects the presence of a significant interaction between the $Rb_2$ molecule and the helium environment, which catalyzes an ultrafast intramolecular relaxation into an electronic state, that has no FC overlap with the initially

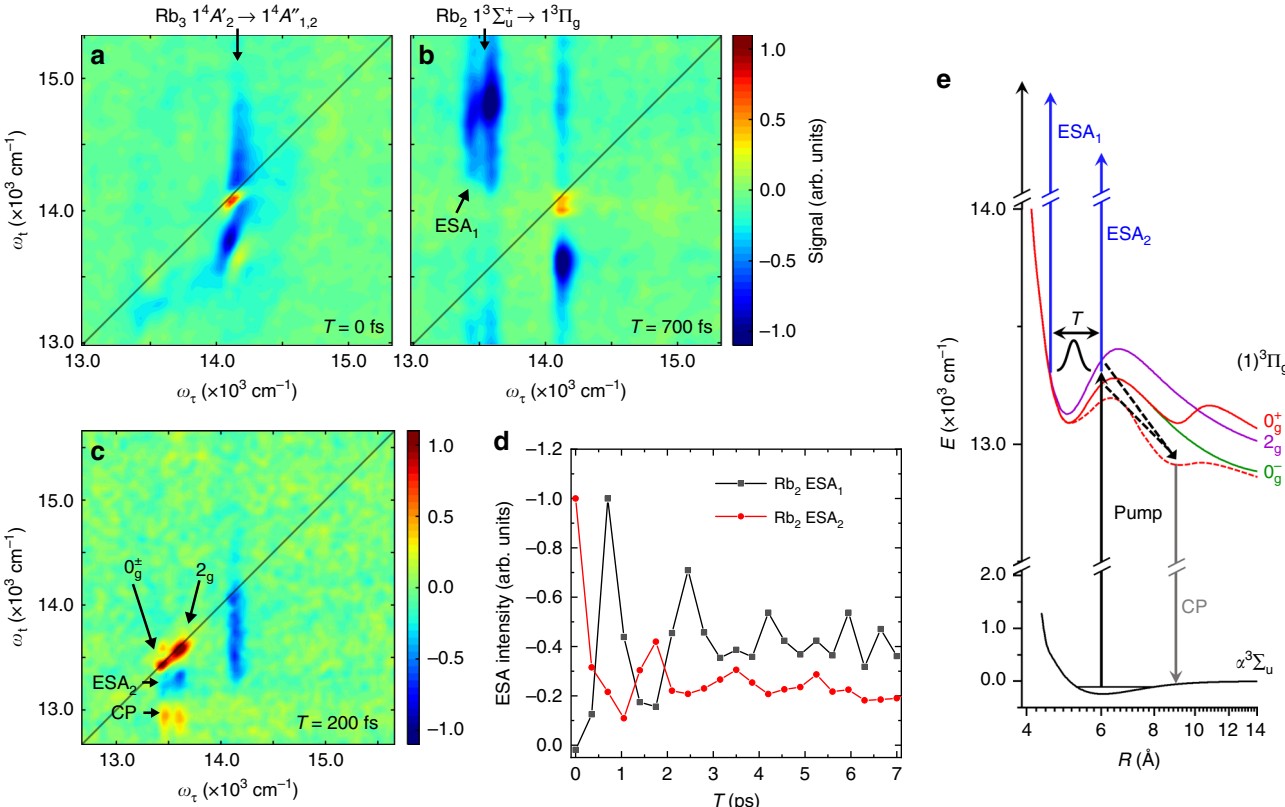

**Fig. 2** Rb$_2$ and Rb$_3$ 2DES results. **a**, **b** Photoelectron-2D correlation spectra of isolated Rb$_2$ and Rb$_3$ molecules for evolution times $T = 0$ fs and 700 fs, respectively. Labels indicate the assigned transitions. **c** Selective enhancement of Rb$_2$ features using a wavelength-optimized fifth pulse combined with photoion detection. **d** Coherent oscillation of Rb$_2$ excited state absorption (ESA) peaks as a function of $T$. **e** Rb$_2$ PECs[28] and concluded photo dynamics. Transitions are labeled in accordance to **b**, **c**. A droplet-induced blue shift of 115 cm$^{-1}$ is applied to the $(1)^3\Pi_g$ states and the influence of the helium perturbation on the $0_g^+$ state is schematically indicated as dashed curve

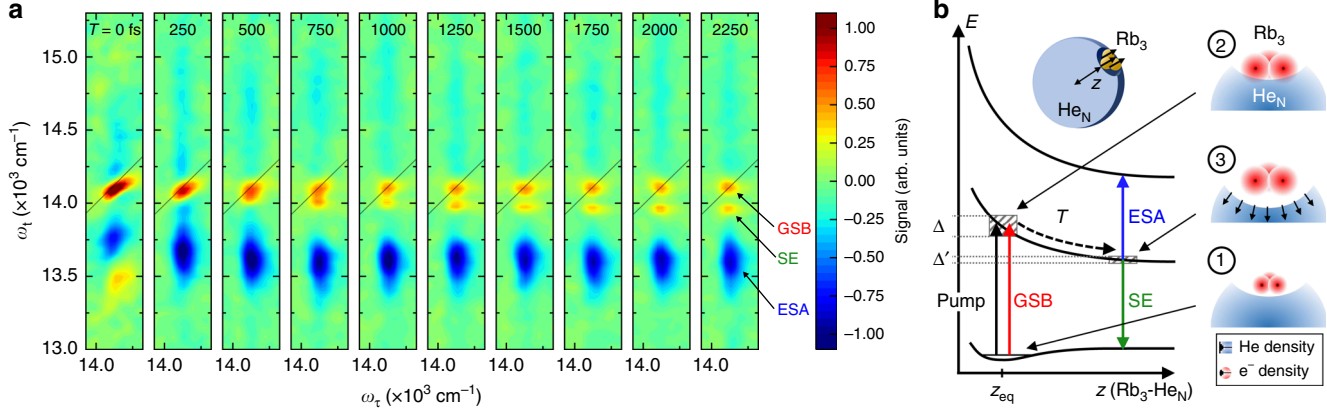

**Fig. 3** Matrix-induced dynamic energy shift of the Rb$_3$ resonance. **a** Time evolution of spectral features correlated to the Rb$_3$ $1^4A_2' \rightarrow 1^4A_{1,2}''$ excitation. For a compact presentation, only a cutout of acquired 2D spectra are shown. **b** Schematic of the Rb$_3$-He$_N$ potentials explaining the matrix-induced dynamic shift. Step 1–3 sketch the expansion of the Rb$_3$ electron orbital upon excitation, followed by a repulsion of the helium density. In stimulated emission (SE) and excited state absorption (ESA) pathways, the system evolves on the excited state during $T$, leading to a dynamic red shift and peak narrowing on the probe axis, which is not present in the ground state bleach (GSB) pathway. $\Delta$, $\Delta'$ indicate the change in line broadening along the $\omega_t$-axis

populated ground state of the cold molecule. Note, no clear oscillation can be resolved in the Rb$_2$ diagonal peaks within our signal quality. This might be due to mainly GSB pathway contributions which propagate on the electronic ground state.

Another example of system-bath interactions is observable in the Rb$_3$ system, where the spectrally isolated quartet resonance $1^4A_2' \rightarrow 1^4A_{1,2}''$ ($\omega_\tau = 14125$ cm$^{-1}$) serves us as a probe for the solvation dynamics of the alkali-metal complex. Here, a second peak emerges from the Rb$_3$ diagonal peak (Fig. 3a), showing a dynamic red shift in the emission frequency, which converges to a constant shift of $\Delta\omega = 150 \pm 19$ cm$^{-1}$ within 2.5 ps (see also Supplementary Fig. 6). The ESA peak below the diagonal shows a similar red shift, however, compromised by an overlapping positive cross peak.

We interpret our data with a dynamic Stokes shift induced by the helium matrix (Fig. 3b). Upon excitation, the molecule's electron distribution expands, causing a repulsion and rearrangement of the surrounding helium density (bubble effect[34]), which leads to an increasing molecule-droplet distance $z$ and thus to a time-dependent reduction of the matrix-induced energy shift. This dynamic is reflected in the SE/ESA pathways as they evolve on the excited state of the $Rb_3$-$He_N$ potential, whereas the GSB pathway evolves on the ground state and shows no time dependence (Supplementary Fig. 3).

At $T = 2.5$ ps, the interaction potential curvature has reached a low gradient, explaining the constant SE/ESA peak position and the reduced line broadening along the emission axis of the SE, whereas at the same time, the GSB contribution exhibits an approximately symmetric peak shape. For $T \geq 50$–$100$ ps, the SE/ESA peaks vanish in contrast to the GSB contribution (not shown) indicating the desorption and accompanied dissociation of the metastable $Rb_3$ quartet molecule. Similar desorption time scales have been deduced for Rb atoms and $Rb_2$ molecules[35,36]. Furthermore, the observed matrix shift of $\Delta\omega = 150 \pm 19$ cm$^{-1}$ for $Rb_3$ is along the line of the shifts for Rb atoms ($\Delta\omega = 12$ cm$^{-1}$)[31] and $Rb_2$ molecules ($\Delta\omega = 115$ cm$^{-1}$, see Supplementary Note 3). Respectively, we deduce for the gas-phase asymptote of the $1^4A_2{}' \rightarrow 1^4A_{1,2}{}''$ transition a value of $13938 \pm 16$ cm$^{-1}$, which was so far unknown at this precision. Likewise, we retrieve temporal information about the rearrangement dynamics of the superfluid helium surface, indicating a time scale <2.5 ps.

Due to strong neighboring cross and ESA peaks flanking the $Rb_2$ diagonal peak (Fig. 2c) and presumably dominating GSB contributions, we were not able to resolve this matrix effect there. We further note, that an excited wave packet passing through a conical intersection may yield a very similar dynamic picture as the one observed. However, this model contradicts available ab initio PECs[30].

## Discussion

While system-bath interactions are not accessible in free gas-phase molecules, HENDI provides an ideal and unique test bench to study molecular dynamics in different environments. Typically, the perturbation of molecular processes by the pure helium droplet is weak[16,17] and environmental effects are modeled by adding impurities to the droplet[23]. However, alkali-metals are an exception and significant couplings even with the pure helium droplet exist[31]. Here, the unshielded valence electrons of alkali-metals cause a repulsive interaction of the helium density (Pauli repulsion) leading to a blue shift and broadening of electronic energies, best described in a pseudo-diatomic molecular model[37] (Fig. 3b). This is in contrast to condensed phase experiments, where interactions are typically attractive and lead to red shifts.

The current work has revealed two peculiar effects arising from this interaction. In the $Rb_2$ molecule, the perturbation by the helium causes a distinct modification of the molecular PECs. This leads to a pronounced cross peak in the 2D spectra (CP feature in Fig. 2c) arising from an ultrafast intramolecular transfer of excited state population into the outer potential well of the $0_g^+$ state, which has so far not been observed in the gas phase.

As a second example of system-bath interactions, a distinct dynamic Stokes shift is observed in the $Rb_3$ molecule, which provides us direct insight into the solvation dynamics of the molecule attached to the helium droplet surface. The behavior of the superfluid helium bath is quite different from typical solvents in condensed phase studies, where solvation effects have been extensively studied[38–43]. There, statistical fluctuations of the solvent induce strong dephasing of electronic coherences which effects the response of single molecules and, in particular, leads to pronounced inhomogeneous broadening in the ensemble average[2]. This is in contrast to the superfluid helium environment, which resembles, due to the highly delocalized helium atoms, an isotropic bath. Droplet-induced perturbations of alkali systems are therefore often well described by a frozen droplet model[31,37], whereas a full treatment including charge density dynamics, typically leads to minor corrections in the guest-host interaction[44]. Hence, inhomogeneous broadening and dephasing of electronic coherences due to statistical bath fluctuations is negligible in these systems. This is confirmed in our study, which provides a direct measurement of the free polarization decay of electronic coherences in molecules attached to helium droplets. We observe a decay within ≈350 fs of electronic coherences (not shown). A Fourier-transform yields a lineshape which is almost identical to the one from previous steady-state absorption spectra of the $Rb_2$ and $Rb_3$ molecules attached to the helium droplet surface (Supplementary Fig. 4). This indicates that the line broadening mechanism is dominated by a static broadening caused by the excitation into the repulsive PECs of the pseudo-molecular $Rb_3$-$He_N$ system (Fig. 3b). At the same time, the dynamic Stokes shift reflects the system's relaxation along the $Rb_3$-$He_N$ interaction coordinate. Due to negligible disorder-induced dephasing, the dynamic Stokes shift is particularly well resolved in our data and has, to the best of our knowledge, never been observed such clearly in 2DES experiments before.

In conclusion, the presented work considerably advances the field of nonlinear spectroscopy in the gas phase by combining coherent multidimensional spectroscopy with helium nanodroplet-assisted synthesis of dilute gaseous nanosystems. In particular, we solved the general issue of insufficient experimental sensitivity and achieved data quality and acquisition times comparable to condensed phase experiments, making gas-phase 2DES studies now feasible for a wide range of applications. As a second key-achievement, our approach has solved the previous strong constraints in the synthesis of gas-phase probes for nonlinear spectroscopy experiments. Our technique uniquely enables the synthesis of isolated nanometer-sized ensembles that feature all essential ingredients of molecular systems, i.e., intramolecular, intermolecular couplings and system-bath interactions. This is in stark contrast to all other gas-phase techniques, where almost exclusively intramolecular couplings are accessible.

The capacity and flexibility of this combination of methods was demonstrated in a model study of high-spin alkali-metal complexes, specifically $Rb_2$ and $Rb_3$. To the best of our knowledge, these experiments represent the first 2DES study of isolated molecules cooled to sub-Kelvin internal temperatures. High-resolution 2D spectra were deduced featuring fine structural details, hard to observe in the condensed phase and we added information about the molecule's femtosecond dynamics, ionization pathways and solvation dynamics within the superfluid helium bath.

The here established gas-phase 2DES approach provides an important extension of condensed phase experiments as coherent molecular dynamics can now be studied in tailored model systems exposed to different environments, for example by co-doping the helium droplets with individual solvent molecules, using other rare gas clusters as substrates or studying free molecules in molecular beams, all readily implemented in our apparatus. In the future, the demonstrated unique flexibility in the synthesis of systems and the high control of experimental conditions will be extremely valuable in answering fundamental questions in primary photophysical and photochemical processes. The achieved high sensitivity will also open 2D spectroscopy studies of other fundamental gas-phase systems, for example mass-selected cluster beams, ultracold quantum gases or ion crystals.

## Methods

**Sample preparation.** Helium nanodroplet beam generation is described in detail elsewhere[45]. An extended sketch of our vacuum apparatus is shown in Supplementary Fig. 1 along with the formation mechanism of Rb molecules. $^4$He gas (purity grade 6.0) is continuously expanded through a nozzle (5 μm diameter) cooled to 17 K with a stagnation pressure of 50 bar, leading to a mean droplet size of 7000 helium atoms. Upon condensation and evaporation, the droplets cool down to 370 mK[16] and undergo a phase transition into the superfluid phase. The droplet beam passes through a temperature-controlled pick-up cell (1 cm length, $T$ = 377 K) containing a low-density Rb vapor ($3.9 \times 10^{-4}$ mbar). Alkali-metal atoms do not immerse into the droplets[31]. Pick-up of multiple atoms thus leads to molecule/cluster formation on the droplet surface. Thereby, the released binding energy is effectively dissipated upon evaporation of helium atoms, assisting the formation of the weakly-bound lowest high-spin electronic states of $Rb_2$ and $Rb_3$ molecules accompanied with cooling to their vibrational ground level. The considerably larger energy release of the low-spin electronic ground states of Rb molecules leads to an enhanced evaporation or droplet destruction, causing a predominance of the high-spin Rb molecules in the experiment[46].

**Optical setup and data acquisition.** The 2DES optical setup is based on the phase modulation (PM) technique developed by Marcus and coworkers[32]. Details are found in the Supplementary Note 1, along with a detailed layout of the optical setup in Supplementary Fig. 2. Briefly, four phase-modulated collinear pulses are focused ($f = 300$ mm) into the interaction region of the detector to induce the nonlinear signals (Fig. 1b). Ionization is either performed with a separate fifth pulse (delayed by $\Delta = 2$ ns) or by absorbing additional photons from pulse 4. Independent wavelength tuning of pulse 5 allows for selective amplification/discrimination of specific spectral features. Photoelectrons or mass-resolved photoions are detected with a channeltron detector. For each evolution time $T$, the coherence times $\tau$ and $t$ are scanned and Fourier transformed afterwards to yield 2D frequency-correlation spectra of which the real part is shown. To isolate the weak nonlinear signal components, the phase $\phi_i$ of each pulse is individually modulated (Fig. 1b) at radio frequencies $\Omega_i$, leading to characteristic modulation signatures for the rephasing ($\Omega_{RP} = \Omega_{43} - \Omega_{21} = 3$ kHz) and non-rephasing ($\Omega_{NRP} = \Omega_{43} + \Omega_{21} = 13$ kHz) third-order signals, which are separated upon lock-in detection. Thereby, amplitude and phase information are retrieved through heterodyned detection by referencing the lock-in amplifier to a suitable reference. The excitation scheme along with double-sided Feynman diagrams and phase signatures of pathway contributions is shown in Supplementary Fig. 3.

## Data availability

The data that support the findings of this study are available from the corresponding author upon reasonable request.

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

## Acknowledgements

We thank A.W. Hauser for fruitful discussions about the Rb$_3$ molecule and W.E. Ernst for providing us the Rb$_2$ and Rb$_3$ laser spectroscopy data. Funding by the European Research Council (ERC) within the Advanced Grant "COCONIS" (694965), by the Deutsche Forschungsgemeinschaft (DFG) IRTG CoCo (2079) and the use of the computing center MésoLUM of LUMAT research federation (FR LUMAT 2764) is acknowledged.

## Author contributions

F.S. and L.B. conceived the experiment. U.B., M.B., D.U., and L.B. implemented the experiment. U.B. and M.B. performed the measurements. O.D., R.V., N.B.-M., and L.B. performed the Rb$_2$ calculations. L.B. wrote the manuscript with input from all other authors.

## Additional information

**Competing interests:** The authors declare no competing interests.

