## [Peer Review File · Nature Communications]

Reviewers' comments:

Reviewer #1 (Remarks to the Author):

This paper describes a nice demonstration experiment that applies a combination of action-signal detected two-dimensional electronic coherence spectroscopy to Rb₂ and Rb₃ molecules, which are formed within a beam of superfluid helium nanodroplets. The work appears to be well carried out, and the results make physical sense and are significant. Such experiments could be useful to understand the coherent excited state dynamics of tailored molecular systems, for which there are very few examples.

I do have a concern regarding the presentation of the article. The action signal-detected 2DES experimental apparatus used in these studies is nearly identical to that developed by others, and even the gas phase and photo-ionization aspects of the measurements were demonstrated previously in a mass spectrometry apparatus by Roeding and Brixner, which was published earlier this year in Nature Communications [doi:10.1038/s41467-018-04927].

The novelty of the current work is in advancing these methods to study coherent excited state dynamics in isolated molecules (and possibly in more complex HENDI prepared systems) in which the system bath interactions can be better controlled than is possible in most experimental systems. This is in itself a significant accomplishment, which deserves the attention of the Nature Communication readership. However, the introduction and discussion sections of the manuscript, in its current form, do not emphasize this aspect of the work over its more technical details, most of which have been described in the literature. The authors should consider revising these sections accordingly to make more clear the significance of their contribution.

Reviewer #3 (Remarks to the Author):

Bruder et al report 2D ES measurements of dilute gas-phase Rubidium molecules. This manuscript reports a method development, not a scientific conclusion. Nevertheless, the measurements are technically impressive and can reveal incisive information about cold, gas-phase small molecules, and therein lies the novelty.

Nearly everything about the measurements is well done, and I find that the work is suitable for this journal.

Minor comments:

(1) On page 7, the authors reference publications dealing with coherent oscillations in photosynthetic proteins. This is distracting and irrelevant, and should be removed. I urge the authors to use the data to learn something deeper about their system, for example comparing dephasing timescale to linewidths to comment on relative contributions to the absorption linewidth or Stokes shift.

(2) Two technical aspects of the measurements are confusing. (A) It was unclear exactly how the authors 'phased' their 2D spectra. (B) Readers would benefit from a short sentence describing how ESA signals appear - such signals have negative amplitude, and since the detection method is based on photoionization, this would imply negative numbers of photoions are detected. Clearly that does not occur, hence my confusion.

(3) The authors observe a beautiful Stokes shift resulting in separated GSB and SE signals. This is unique, and therefore I urge the authors to use this opportunity to explore the solvation dynamics of their system. Of course "solvation" is a curious word in this context, but nevertheless the authors could make contact with the extensive literature of solvation dynamics in condensed phase systems. See "Broadband pump-probe spectroscopy quantifies ultrafast solvation dynamics of proteins and molecules" JPC Lett 7, 4722-4731 (2016), for example.

D. Turner

Reply to the reviewers' comments

We thank the reviewers for their positive evaluation of our manuscript and their suggestion for publication in Nature Communications. The reviewers' suggestion for changes in the presentation of our manuscript are addressed below.

The reviewers' original comments are in black, our answer is in blue. Changes in the manuscript are also marked in blue.

Reviewer #1 (Remarks to the Author):

This paper describes a nice demonstration experiment that applies a combination of action-signal detected two-dimensional electronic coherence spectroscopy to Rb₂ and Rb₃ molecules, which are formed within a beam of superfluid helium nanodroplets. The work appears to be well carried out, and the results make physical sense and are significant. Such experiments could be useful to understand the coherent excited state dynamics of tailored molecular systems, for which there are very few examples.

I do have a concern regarding the presentation of the article. The action signal-detected 2DES experimental apparatus used in these studies is nearly identical to that developed by others, and even the gas phase and photo-ionization aspects of the measurements were demonstrated previously in a mass spectrometry apparatus by Roeding and Brixner, which was published earlier this year in Nature Communications [doi:10.1038/s41467-018-04927].

The novelty of the current work is in advancing these methods to study coherent excited state dynamics in isolated molecules (and possibly in more complex HENDI prepared systems) in which the system bath interactions can be better controlled than is possible in most experimental systems. This is in itself a significant accomplishment, which deserves the attention of the Nature Communication readership. However, the introduction and discussion sections of the manuscript, in its current form, do not emphasize this aspect of the work over its more technical details, most of which have been described in the literature. The authors should consider revising these sections accordingly to make more clear the significance of their contribution.

We thank the reviewer for the valuable comments. We have rewritten the introduction and conclusion of our manuscript to address the reviewer's point (See changes in manuscript on p. 2-4 and p. 13-14). However, we refrained from shortening background information as the journal addresses a broad readership which may not be familiar with the employed techniques. In addition, a summary of the advances achieved in our work, clearly extending the mentioned previous work, is given below:

Our work solves the long-standing issue of insufficient sensitivity for gas-phase 2DES experiments: The data quality and required acquisition times in the work of Röding and Brixner stand far behind the level of condensed phase 2DES experiments. In contrast, we achieve a huge improvement in sensitivity (4 orders of magnitude less sample density in our experiment). At the same time, our data quality and acquisition times are now comparable with condensed phase experiments which makes gas-phase 2DES now feasible for a wide range of applications.

Furthermore, we incorporate a more flexible photoionization scheme using a tunable ionization laser. With this, we demonstrate selective amplification of weak spectroscopic features which provides an additional means to disentangle the nonlinear response of the system.

Advanced experimental setup:

In the work of Röding and Brixner, a very simplified experimental apparatus is used, consisting of a single vacuum chamber, capable of only producing thermal molecular gases at the upper limit of tolerable vacuum conditions for detection. Our vacuum apparatus and sample preparation is greatly advanced to their setup. In a fully elaborated molecular beam machine, multiply differentially pumped at UHV conditions, we apply 2DES for the first time to a supersonic molecular/cluster beam that prepares molecular samples at sub-Kelvin internal temperatures. This demonstrates the full power of gas-phase studies, enabling us to record high quality 2D data with unprecedented clear details.

Our work solves the issue of the strongly constraint flexibility in the synthesis of gas-phase samples for nonlinear spectroscopy:

Helium nanodroplet isolation provides a fundamental conceptual advantage over thermal molecular gases or molecular beams, as it uniquely facilitates the synthesis of isolated model systems which feature all essential ingredients of a quantum system, i.e. millikelvin temperature, intra- and inter-particle interactions (molecular aggregates/complexes) and the coupling to a controllable environment (helium droplet, microsolvation by co-doping).

Briefly summarized:

- First time: 2DES of isolated, cold molecules, state-selectively prepared at millikelvin temperatures.
- New level of sensitivity: Facilitating 2DES of extremely dilute probes (10^7 cm^{-3} , 10^{-11} optical density) and introducing photoionization as means of selective probing.
- New level of resolution in 2D spectra: Revealing features not observed at all, or in such detail in previous studies (Spin-orbit effects, Stokes shift, perturbation effects).
- First time: High-spin alkali trimer solvation in the superfluid helium surface revealed in real-time and quantified to new precision.

Reviewer #3 (Remarks to the Author):

Bruder et al report 2D ES measurements of dilute gas-phase Rubidium molecules. This manuscript reports a method development, not a scientific conclusion. Nevertheless, the measurements are technically impressive and can reveal incisive information about cold, gas-phase small molecules, and therein lies the novelty.

Nearly everything about the measurements is well done, and I find that the work is suitable for this journal.

We point out that our method is indeed not limited to the investigation of small molecules. The Rb molecules serve here rather as clean model system to demonstrate the principle of our novel experimental approach.

Minor comments:

(1) On page 7, the authors reference publications dealing with coherent oscillations in photosynthetic proteins. This is distracting and irrelevant, and should be removed.

We agree that this reference to other work might be distracting and have changed the manuscript accordingly p. 8, para. 2.

I urge the authors to use the data to learn something deeper about their system, for example comparing dephasing timescale to linewidths to comment on relative contributions to the absorption linewidth or Stokes shift.

We included this in our extended discussion which has been added to the manuscript (see answer to point (3)).

(2) Two technical aspects of the measurements are confusing. (A) It was unclear exactly how the authors 'phased' their 2D spectra. (B) Readers would benefit from a short sentence describing how ESA signals appear - such signals have negative amplitude, and since the detection method is based on photoionization, this would imply negative numbers of photoions are detected. Clearly that does not occur, hence my confusion.

Concerning the phasing we have added the following text in the SI (p. 3, para. 2) to clarify this point:

“Correct phasing of the 2D spectra is crucial and often a technical issue in 2DES⁵². As an advantage of the phase modulation approach, phasing is readily done by adjusting a reference phase in the lock-in amplifier as described in Ref. 29. In our photoionization experiments we cross-checked the correct phasing with 2DES measurements of atomic Rb in an effusive atomic beam which provides us simplified, particularly sharp 2D spectra of which the phase behavior is well-known. “

For a better explanation of negative phases we have added the following in the SI (p.2, para. 4):

“The absolute magnitude of these signals reflects the detected photoelectron/-ion count rates contributing to the individual pathways. However, the phase of each feature arises from heterodyned detection of the nonlinear signals with a reference waveform, which is done in the lock-in amplifier. This leads to a positive amplitude of SE/GSB pathways and a negative amplitude of ESA pathways as shown in Fig. S3.“

(3) The authors observe a beautiful Stokes shift resulting in separated GSB and SE signals. This is unique, and therefore I urge the authors to use this opportunity to explore the solvation dynamics of their system. Of course "solvation" is a curious word in this context, but nevertheless the authors could make contact with the extensive literature of solvation dynamics in condensed phase systems. See "Broadband pump-probe spectroscopy quantifies ultrafast solvation dynamics of proteins and molecules" JPC Lett 7, 4722-4731 (2016), for example.

We thank the reviewer for this valuable comment and have extended our discussion of this aspect accordingly in our manuscript. You find now a new discussion section (p. 11-13), discussing point (1) and (3). The main points added are:

The superfluid helium environment provides an isotropic environment with negligible statistical fluctuations. Therefore, pure dephasing and inhomogeneous broadening is hardly present which is in strong contrast to most condensed phase experiments. Our experiment provides the first measurement of a free induction decay of electronic coherences in molecules attached to a helium droplet. A comparison with steady-state spectroscopy confirms the negligible inhomogeneity and disorder-induced dephasing. The line broadening is thus mainly determined by static broadening described by repulsive potential energy curves in a pseudo-diatom molecular model.

REVIEWERS' COMMENTS:

Reviewer #1 (Remarks to the Author):

This reviewer provided confidential remarks to the editors recommending publication.

Reviewer #3 (Remarks to the Author):

The authors have responded satisfactorily to my previous queries as well as those of the other reviewer. The authors have made their technical advances in the method development more clear to readers, and the Discussion section focused on solvation increases the breadth and depth of the manuscript.

One minor comment is that the authors use the phrase "free induction decay". While the intent might be clear, the physical mechanism does not apply to these measurements. Induction applies to the decay of the magnetization in an NMR measurement. This should be adjusted to "free polarization decay", which conveys the physical mechanism in action in these measurements.

Reply to the reviewers' comments

Reviewer #1 (Remarks to the Author):

This reviewer provided confidential remarks to the editors recommending publication.

Reviewer #3 (Remarks to the Author):

The authors have responded satisfactorily to my previous queries as well as those of the other reviewer. The authors have made their technical advances in the method development more clear to readers, and the Discussion section focused on solvation increases the breadth and depth of the manuscript.

One minor comment is that the authors use the phrase "free induction decay". While the intent might be clear, the physical mechanism is does not apply to these measurements. Induction applies to the decay of the magnetization in an NMR measurement. This should be adjusted to "free polarization decay", which conveys the physical mechanism in action in these measurements.

We agree with the reviewer that, although broadly used in IR and VIS spectroscopy, the term "free induction decay" is strictly speaking not correct in the used. As suggested, we changed the wording in the manuscript (p. 12, para. 3) to "free polarization decay".